# TALEN-Mediated Gene Editing of *slc24a5* (Solute Carrier Family 24, Member 5) in Kawakawa, *Euthynnus affinis*

**Dipak Pandey [1,*], Takahiro Matsubara [1], Taiju Saito [1], Yukinori Kazeto [2], Koichiro Gen [3], Tetsushi Sakuma [4], Takashi Yamamoto [4], Miyuki Mekuchi [5] and Rie Goto [1,*]**

[1] South Ehime Fisheries Research Center, Ehime University, 25-1 Ainan-cho, Minami-Uwa, Ehime 798-4206, Japan; matsubara.takahiro.mc@ehime-u.ac.jp (T.M.); saito.taiju.bk@ehime-u.ac.jp (T.S.)

[2] Minamiizu Field Station, Fisheries Technology Institute, Japan Fisheries Research and Education Agency, 183-2 Minamiizu, Kamo, Shizuoka 415-0156, Japan; kazeto@fra.affrc.go.jp

[3] Nagasaki Field Station, Fisheries Technology Institute, Japan Fisheries Research and Education Agency, 1551-8 Taira-machi, Nagasaki 851-2213, Japan; kgen@affrc.go.jp

[4] Graduate School of Integrated Sciences for Life, Hiroshima University, 1-3-1 Kagamiyama, Higashi-Hiroshima 739-8526, Japan; tetsushi-sakuma@hiroshima-u.ac.jp (T.S.); tybig@hiroshima-u.ac.jp (T.Y.)

[5] Yokohama Field Station, Fisheries Resources Institute, Japan Fisheries Research and Education Agency, 2-12-4 Hukuura, Kanazawa-ku, Yokohama 236-8648, Japan; mekuchim@affrc.go.jp

[*] Correspondence: pandey.dipak.ww@ehime-u.ac.jp (D.P.); goto.rie.me@ehime-u.ac.jp (R.G.)

**Abstract:** Transcription activator-like effector (TALE) nucleases (TALENs) mediated gene editing methods are becoming popular and have revealed the staggering complexity of genome control during development. Here, we present a simple and efficient gene knockout using TALENs in kawakawa, *Euthynnus affinis*, using *slc24a5*. We examined *slc24a5* gene expression and functional differences between two TALENs that hold the TALE scaffolds, +153/+47 and +136/+63 and target *slc24a5*. Developmental changes in *slc24a5* transcripts were seen in early-stage embryos by real-time PCR; *slc24a5* expression was first detected 48 h post fertilization (hpf), which increased dramatically at 72 hpf. Four TALENs, 47- and 63-type of two different target loci (A and B), respectively, were constructed using Platinum TALEN and evaluated in vitro by a single-strand annealing (SSA) assay. TALEN activities were further evaluated in vivo by injecting TALEN mRNAs in the two-cell stage of the zygote. Most of the TALEN-induced mutants showed mosaic patterns in the retinal pigment epithelium (RPE) and fewer melanin pigments on the body at 72 hpf and later when compared to the control, implying the gene's association with melanin pigment formation. A heteroduplex mobility assay (HMA) and the genome sequence further confirmed the TALEN-induced mutations of substitution, insertion, and deletion at an endogenous locus.

**Keywords:** kawakawa; *slc24a5*; TALEN; melanin; SSA; HMA

## 1. Introduction

Kawakawa (*Euthynnus affinis*), also known as mackerel tuna, is an Indo-Pacific species belonging to the family Scombridae and forms an important commercial fishery in Southeast Asia. It is closely related to the bluefin tuna [1]. This fish inhabits tropical and subtropical waters with seawater temperature ranging from 18 to 29 °C, in the coastal region of southeastern Japan. The Kuroshio current marks the northern limit of its habitat.

Although the kawakawa fishery in Japan relies mainly on wild-caught fish, there has been recent interest in Japan in the aquaculture of this species as it is considered a delicacy, has a body size that makes it suitable for aquaculture (30–60 cm; 2.5 ± 0.5 kg), and reaches market size in a relatively short period [2]. Kawakawa has also received much attention as a useful model organism for studying the genetic basis of growth and swimming speed of bluefin tuna. Successful aquaculture of this fish will help further this research as well.

Aquaculture of kawakawa will also aid in ensuring the conservation of wild stocks. Planning for closed-cycle aquaculture of this species with the help of technologies such as genomics will further aid in conserving the wild stocks, while helping to ensure sustainable aquaculture production as well [3].

Perhaps the most useful genome-based technologies are in the area of genome editing, which have revolutionized the field [4]. Recent genome editing technologies have found extensive applications in different fish species for basic functional studies as well as in applied research in the field of aquaculture [4–12]. In particular, genome editing can also be used to produce desirable fish breeds [10]. There are three important gene editing technologies that have revolutionized the field and are based on zinc-finger nucleases (ZFNs), transcription activator-like effector (TALE) nucleases (TALENs), and clustered regularly interspaced short palindromic repeats (CRISPR)/CRISPR-associated protein 9 (Cas9) (CRISPR/Cas9). Although each has its own set of pros and cons, the specificity and efficiency, with low off-target effects, are known to be comparatively higher for TALENs and CRISPR/Cas9 [4,13,14]. TALEN is a second-generation genome editing tool consisting of the TALE protein containing customizable DNA-binding repeats and the nuclease domain of FokI enzyme. However, there has been a complementary usage of these two technologies in recent years, where CRISPR/Cas9 works as monomer, consisting of protein and RNA, producing blunt ends, while TALEN works as dimer, contains protein only and produces cohesive ends [15]. However, due to the long recognition site of TALEN technology (30–40 bp) as compared to short recognition site of CRISPR/Cas9 (only around 22 bp), TALENs have a higher specificity, with a lower probability of introducing off-targets into the genome.

An artificial site-specific nuclease such as TALEN offers powerful tools for genome editing by enabling gene disruption by mutation [16]. It was originally discovered in the bacterial plant pathogen *Xanthomonas* in the form of the proteins, TAL effectors (TALEs) [13,17]. TALENs consist of a TALE domain containing a set of RVDs (repeat-variable diresidues) linked to a *FokI* monomer and can be used to make targeted DNA double-strand breaks (DSBs) to introduce mutations at precise locations in the genome. The DSBs are repaired through non-homologous end joining (NHEJ), which directly connects the ends of the broken strands to efficiently introduce insertion and/or deletion (indel) mutations. Indels can lead to frameshift mutations in the targeted gene, which often result in the loss of function of the encoded protein. TALENs have been used for genome editing in various cells and organisms, including frogs [18] and fishes [5,17–26]. Overall reports showed the TALEN-mediated genome editing technology can precisely modify predetermined loci in plants and animals, that make it possible to optimize plant and animal production. Thus, potential of TALENs technology improve aquaculture industries.

As genome editing of genes associated with body color has been successfully used in zebrafish [27,28], Atlantic salmon [29], Chinese lamprey [7], frog [30], and common carp [11], this technique can be used to ascertain the efficacy of the technique in novel applications. Pigmentation pattern is one of the most diverse phenotypic traits in fishes [31]. In fishes and other vertebrates, cells that determine pigmentation patterns originate from neural crest cells during embryogenesis and then migrate to their specific locations to form the final pigmentation pattern [32–34]. Fish exhibit a common dorsal–ventral countershading pigment pattern that is formed by the differential distribution of multiple types of chromatophores, which present diverse color phenotypes with the dark dorsal skin and pale belly. An important gene that regulates the appropriate melanin synthesis is solute carrier family 24, member 5 *(slc24a5)*, also known as sodium/potassium/calcium exchanger 5 (NCKX5) [35]. It was first cloned in zebrafish and confirmed to control the golden phenotype [36], where the *slc24a5* protein functions in intracellular membrane-bound structures in a manner consistent with melanosomes and/or their precursors that when mutated, lead to lighter pigmentation such as a golden phenotype. The golden phenotype is characterized by hypopigmentation of skin melanophores and retinal pigment epithelium 'RPE' [37]. Structurally, melanin has been defined in several ways during the

last 50 years [38–40]. Melanin produced by cells known as melanocytes, which are a type of biopolymer, is made up of a complex chemical compound (indole-5, 6-quinone) produced from tyrosine [41]. Genes that can influence the melanin deposition are involved in many biological pathways, such as melanocyte development, components of melanosomes and their precursors, melanosome construction/protein routing, melanosome transport, and systemic effects of eumelanin and pheomelanin [42]. The development of a pigment cell is a complex process, associated with a series of cellular, genetic, nutritional, and physiological factors [43]. In addition, several environmental factors, especially those associated with improper nutrition, have been shown to introduce pigmentation problems. Perhaps due to the ubiquitous role of pigmentation in affecting morphological diversity, researchers have sought to understand how pigmentation patterns are built during animal development [44].

Currently, genome editing of *slc24a5* is only available for freshwater model species such as zebrafish [36] and frog [30]. In this study, we examine whether the genome editing technique can be applied to kawakawa as well. For this purpose, we selected *slc24a5* as the target gene for genome editing using Platinum Gate TALEN kit (Addgene: Kit #1000000043). Platinum TALENs have variable TALE repeats with one of two types of TALE scaffold: the DSB-forming activities of TALENs with the +153/+47 (47-type) scaffold that are restricted to shorter spacers, and scaffold +136/+63 (63-type), which are capable of inducing DSBs in a wide range of spacer lengths [45], although both types of TALEN scaffolds have high levels of activity [45]. We produced 47-type and 63-type scaffolds at the same target sequences of kawakawa *slc24a5* in flounders and their in vivo activity by examining phenotype change, genome modification, and survival rate.

## 2. Materials and Methods

### 2.1. Experimental Animals and Tissue Collection

Several hundred juvenile wild kawakawa were captured in the Uwa Sea of Japan in 2016 and kept in a sea cage until use. In 2017, three females and three males of adult kawakawa were sacrificed and tissue collected for analysis as described below. The remaining kawakawa (now adult) were kept in the sea cage for spawning. Details are described in the sub-sections below.

All experimental animals were handled in accordance with the guidelines of South Ehime Fisheries Research Center, Ehime University for the use and care of experimental animals. These fish were genetically modified by means of transgenic technology and handled inside our fish facility with institutional permission (permission number 'AIDAI nansuise 000001'). Fertilized eggs used in the experiments were obtained from the sea cage using the protocol of Goto et al. (2019).

### 2.2. Cloning of cDNA Encoding Putative Kawakawa slc24a5

*slc24a5* cDNA was isolated by extracting total RNA from retinal tissue using RNeasy Plus Universal Mini Kit (QIAGEN, Tokyo, Japan), according to manufacturer's instructions. Poly (A) RNA was purified with Oligotex-dT30 <SUPER> (Takara Bio, Otsu, Japan) and reverse transcribed to construct a RACE cDNA library using the SMARTer RACE cDNA Amplification Kit (TAKARA Bio., Japan). A fragment of *slc24a5* was amplified from retinal RACE cDNA library using a set of tuna specific primers, t_F1 and t_R1 (Table 1), designed based on the sequence of *slc24a5* from Pacific bluefin tuna, obtained from the Pacific bluefin tuna genome database at National Research Institute of Fisheries Science, Japan Fisheries Research and Education Agency. Due to low transcript abundance, a nested PCR was carried out using another set of tuna specific primers t_F2 and t_R2 (Table 1). A sequence of approximately 720 bp from the PCR product was inserted into a *pGEM®-T Easy* vector (Promega, Japan) and sequenced. Subsequently, 5'- and 3'-RACE of retinal tissue cDNA was used to amplify the sequence of the 5' and 3' ends of kawakawa cDNA according to the manufacturer's instructions. The first 5'- and 3'-RACE PCR was done

using the gene-specific primers, 5′RACE_R1 and 3′RACE_F1, followed by a second set of RACE PCRs using primers 5′RACE_Rn and 3′RACE_Fn, respectively. The amplicons including the entire open reading frames for the *slc24a5* were isolated with the specific primers, ORF_F and ORF_R (Table 1), using KOD FX Neo polymerase (TOYOBO, Osaka, Japan) and sequenced. A complete cDNA sequence encoding *slc24a5* was manually assembled using software CLC Sequence Viewer 8.0. The sequences were aligned using ClustalW. The neighbor-joining phylogenetic tree was constructed using CLC Sequence Viewer version 8. The GenBank accession numbers of the *slc24a5* sequences used in this study are shown in Figure S2.

**Table 1.** List of primers used for genome editing of *slc24a5* in kawakawa, *Euthynnus affinis.*

| Name | Direction | Nucleotide Sequence | TM | Use and Its Target |
|------|-----------|---------------------|-----|--------------------|
| t_F1 | Sense | 5′-GAGTTTCCTGACGGCTTCTTC-3′ | 58 | bluefin tuna specific for *slc24a5* |
| t_F2 | Sense | 5′-CAGGAGCAACATTTATGGCAG-3′ | 56 | bluefin tuna specific for *slc24a5* |
| t_R1 | Anti-sense | 5′-CACCGACAACAGTGACCATC-3′ | 57 | bluefin tuna specific for *slc24a5* |
| t_R2 | Anti-sense | 5′-GCAGGCTGAGTGAGAGATGAG-3′ | 60 | bluefin tuna specific for *slc24a5* |
| 5′RACE_R1 | Anti-sense | 5′-AGTACCCCGAGTCGTCTTGGAAGATC-3′ | 63 | 5′-Race for *slc24a5* cDNA |
| 5′RACE_Rn | Anti-sense | 5′-CTGGCAGCTTCATACCAATACACCTTG-3′ | 62 | 5′-Race for *slc24a5* cDNA |
| 3′RACE_F1 | Sense | 5′-GCTTATTTTCTACATGCTTTTGGCTGTCG-3′ | 63 | 3′-Race for *slc24a5* cDNA |
| 3′RACE_Fn | Sense | 5′-GTGGGGTCAGCAGTCTACAACCTCTTAG-3′ | 64 | 3′-Race for *slc24a5* cDNA |
| ORF_F | Sense | 5′-GCATGCTTGCCTATGATTCC-3′ | 57 | ORF of *slc24a5* cDNA |
| ORF_R | Anti-sense | 5′-GTCCTGGGAAGGCAAGTTTC-3′ | 56 | ORF of *slc24a5* cDNA |
| TD_F | Sense | 5′-GAGTTTCCTGACGGCTTCTTC-3′ | 59 | RT-PCR of *slc24a5* |
| TD_R | Anti-sense | 5′-GCGAAGGTCAAAGCACAGAAC-3′ | 60 | RT-PCR of *slc24a5* |
| genome_F | Sense | 5′-CACACAGCTTCAAGCCACAT-3′ | 55 | exon1 of *slc24a5* |
| genome_R | Anti-sense | 5′-AGAAGCCGTCAGGAAACTCA-3′ | 55 | exon1 of *slc24a5* |
| HMA_F | Sense | 5′-CTTTGCATGCTTGCCTATGA-3′ | 58 | RT-PCR, HMA assay |
| HMA_R | Anti-sense | 5′-TGACAGAGTGGGTTTCCTGA-3′ | 58 | RT-PCR, HMA assay |
| qPCR_F | Sense | 5′-GCTGGGAGTTCTGCACCTGA-3′ | 60 | qPCR of *slc24a5* |
| qPCR_R | Anti-sense | 5′-GCCATAGAGGCCAAGAGTCC-3′ | 60 | qPCR of *slc24a5* |
| β-act_F1 | Sense | 5′-CAGGCATCAGGGAGTGATGG-3′ | 57 | primer for internal control |
| β-act_R1 | Anti-sense | 5′-GCTCGATGGGGTACTTCAGG-3′ | 58 | primer for internal control |

*2.3. RT-PCR Analysis of Kawakawa slc24a5 Transcripts in Adult Tissues and Early Stages of Embryos*

For tissue-specific expression of the *slc24a5* transcript, brain, gonad, gills, liver, pylorus, gut, head kidney, lateral kidney, white muscle, red muscle, heart, skin, and retinal tissue were collected from three adult females and males and put into RNAlater and stored at −80℃. Isogene (Nippon-GENE, Tokyo, Japan) was used to isolate total RNA from these samples, and 500 ng of total RNA was reverse-transcribed using the cDNA synthesis kit ReverTraACE with gDNA remover (TOYOBO, Osaka, Japan). Fertilized eggs were placed in 1L seawater with 0.1% penicillin and streptomycin and incubated at 24 ± 0.5 °C to analyze *slc24a5* RNA expression during early development. Five embryos and larvae each were collected at the 2-cell, 128-cell, 1k, 50% epiboly, early-somite at 15 h post fertilization (hpf), late-somite at 24 hpf, 2 days post fertilization (dpf), 3 dpf, 5 dpf, and 10 dpf. The samples were placed in DNA/RNA shield™ (ZYMO RESEARCH, CA, USA) and stored at -80 °C. RNA was isolated from these samples using ZR-*Duct*™ DNA/RNA Mini-iPrep Plus (ZYMO RESEARCH), according to the manufacturer's protocol. Next, 60 ng of total RNA from each tissue at each developmental stage of the embryos and larvae was reverse transcribed using cDNA synthesis kit ReverTraACE with gDNA remover. The *slc24a5* RNA expression in each tissue and embryo and larval stage was examined using a set of *slc24a5* specific primers, TD_F and TD_R (Table 1) and the *β-actin* primers, β-act_F1 and β-act_R1, as the internal control. RT-PCR was carried out with the following

conditions: pre-cycling hold at 95 °C for 2 min, 32 cycles of 94 °C for 30 s, 58 °C for 30 s, 72 °C for 30 s. Next, 10 μL of each PCR product together with 1 μL post-stain dye (Midori Green direct; Nippon Genetics) was run on a 2% agarose gel at 100 V for 25 min. The gel image was captured using iBright™ Imaging System (iBrightFL100, Invitrogen, CA, USA).

*2.4. Quantitative Real-Time PCR (qPCR)*

A single embryo from each of various developmental stages (13, 24, 72, and 120 hpf) from TALEN induced mutants and control groups were placed in 100 μL of DNA/RNA shield™ and stored at −80 °C. RNA was isolated from these samples using ZR-*Duct*™ DNA/RNA MiniPrep Plus, according to the manufacturer's protocol. Next, 60 ng of total RNA from each embryo was reverse transcribed using cDNA synthesis kit ReverTraACE with gDNA remover. A set of real-time primers qPCR_F and qPCR_R (for *slc24a5* transcript), and β-act_F1 and β-act_R1 (Table 1) were designed based on the handbook (CFX96™ Real-Time PCR Detection System). Quantitative real-time PCR was done using SsoFast™ EvaGreen® Supermix on CFX96™ Real-Time PCR Detection System (Bio-Rad, Japan), according to the manufacturer's instructions. PCR was done with the following conditions: pre-cycling hold at 95 °C for 1 min, 40 cycles of 95 °C for 5 s and 60 °C for 15 s. $C_T$ method was used for quantity calculations. The quantitative real-time data was collected and a histogram of the absolute RNA levels (copy number) to starting amount of RNA (per ng RNA) was drawn.

*2.5. Genome Sequence and TALEN Design for Kawakawa slc24a5*

Whole genome sequencing and de novo assembly of kawakawa was recently done in the South Ehime Fisheries Research Center (SEFRC), Ehime University [46]. The full length kawakawa *slc24a5* (ka-*slc24a5*) gene sequence was extracted from the draft of the whole genome sequence (can be found below; doi: https://www.ddbj.nig.ac.jp/,03 December 2021, BPLY01000001-BPLY01010237, ICRR01000001-ICRR01049510, DRA012118, DRA012140, DRA012141) to determine its genome location. The TALEN target sites of ka-*slc24a5* are located in exon 1 (Figure 1). To avoid selecting single nucleotides (SNPs) for the TALEN target sites, exon 1 genome sequence of ka-*slc24a5* was examined from five individuals of wild kawakawa that were captured from the Uwa Sea of Japan. Briefly, genomic DNA was extracted from the fin using DNeasy Blood and Tissue Kits (Qiagen, Tokyo, Japan). Genomic fragments containing exon 1 of ka-*slc24a5* were amplified using specific primers (genome_F and genome_R) (Table 1) and KOD FX Neo polymerase and sequenced. Next, potential TALEN target sites in exon 1 of the ka-*slc24a5* gene using the TALEN Targeter program were scanned [47]. Two TALEN targets were identified and named ka-*slc24a5*-A and ka-*slc24a5*-B, respectively (Figure 1b). The ka-*slc24a5*-A site comprised 19 bp of the left binding site (5′-TCTGCAGAAAAAGAAAAGA-3′), 18 bp of the right binding site (5′-TACTTCCTGGGATTTGTA -3′), and a 15 bp spacer sequence (5′-AAAGATTTTATTCCT-3′). Similarly, the ka-*slc24a5*-B was made up of 19 bp of the left binding site (5′-TCCTGGGATTTGTACTATT-3′), 18 bp of right binding site (5′-CATCTT-GTATCATTCACA-3′), and 15 bp spacer sequence (5′-TTTATATTGCACRGYC-3′) (Figure 1b). Two types of scaffolds, +157/+47 and +136/+63, respectively, were constructed at these two loci to compare the in vivo activities of TALENs (hereinafter referred to as ka-*slc24a5*-47A, ka-*slc24a5*-63A, ka-*slc24a5*-47B and ka-*slc24a5*-63B). The activity of the constructed Platinum TALEN plasmids was first confirmed in vitro using a single-strand annealing (SSA) assay [48]. Briefly, each TALENs activity was compared with cells transfected with TALENs and the reporter vector bearing unrelated target sequences, cells transfected with TALENs and the associated SSA reporter. The TALEN mRNAs were then prepared using mMESSAGE mMACHINE T7 Ultra Kit (Life Technologies, USA), according to manufacturer's instructions and stored at −80 °C until use.

a

slc24a5 gene of kawakawa (5.7kb)

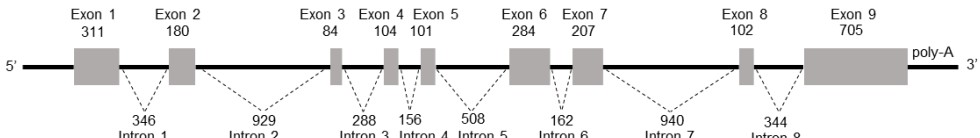

b

>Exon 1 (Partial)

```
TT.TTTACCTTTGCATGCTTGCCTATGATTCCTTAGAATAATTCCTTCTTCTTYTTTTY.......
...TTTTTTTACTCACCATGAATAAGGTTGCAGCTCTGCAGAAAAAGAAAAGAAAAGATTTTATTC
CTTACTTCCTGGGATTTGTACTATTTTTATATTGCACRGYCCATCTTGTATCATTCACAGCAAAAA
CAACTCAGGAAACCCACTCTGTCAGGGTGCGCCGGGYCTTG
```

[..]-nucleotide gap; R=A/G; Y=C/T; [**X**]-start codon;

Target A; grey shaded , Target B; underlined.

**Figure 1.** Design of *slc24a5* TALENs in kawakawa, *Euthynnus affinis*. (**a**) Schematic representation of the structure of the complete sequence of the *slc24a5 gene*. Coding and untranslated exons regions are shown as solid and open boxes, respectively. The number of nucleotides present in each exon and intron is shown. (**b**) *slc24a5* TALENs designed to target exon1 of the gene. The exon1 sequence was determined based on the genomes of five different individuals. The grey shaded and underlined sequences indicate the TALEN binding sites of targets A and B, respectively.

### 2.6. Microinjection of ka-slc24a5 TALENs

Microinjection of *slc24a5*-TALENs was done using a needle with an added constriction (3–5 μm inner diameter) in the upper part of the tip in order to control the pressure required for injecting the RNA solution into the embryo and to prevent backflow of cytoplasm from the embryo as described previously [49]. Embryos at the 1–2 cell stage were placed on a 1% agar-coated Petri dish filled with 50% filtered seawater, and then microinjected with TALEN mRNA at different doses (50, 100, and 150 ng) each of the left and right TALENs as preliminary experiment to confirm the RNA efficiency. Finally, we confirmed final concentration of 150 ng/μL each of left and right TALENs has high efficiency for either group and used the same dose throughout the experiment. The injected embryos were cultured in a 1 L beaker with 800 mL filtered seawater containing antibiotics (0.1% penicillin and streptomycin) at 24 ± 0.5 °C incubator. Survival and developmental rate of embryos were determined at 12 hpf. Embryos were placed in a 24-well plate with a single embryo per well and cultured for periodical observation at 13, 15, 24, 30, 48, 72, 96, 120, and 240 hpf under the stereomicroscope 'SZX16' equipped with DP73 CCD camera (Olympus, Japan). Based on phenotype, the specimens were classified into Types I, II, and III, according to pigmentation pattern. Five hatchlings from each TALEN and five from the control were randomly chosen for further analysis. Deformed embryos were categorized as a Type IV phenotype at 72 hpf. At 120 hpf, they were placed in 100 % EtOH for genotyping. The rest of the specimens were placed in a 1000L circular tank for rearing. Juveniles were reared at 24 ± 0.5 °C and fed a diet of rotifer from 2 dph until 10 dph. They were then fed live hatchlings of striped beakfish and Japanese whiting along with chlorella and rotifer until 17 dph, and subsequently, minced frozen bait until analysis.

### 2.7. Genotyping TALEN-Induced Mutations

Genomic DNA was extracted from 5 dpf larvae using DNeasy Blood and Tissue Kits, according to manufacturer's instructions. A primer set, HMA-F and HMA_R (Table 1), for

genotyping of TALEN induced mutations was designed to amplify the fragment containing both target sites, A and B in exon-I. PCR using HMA_F and HMA_R, and KOD FX NEO polymerase was done as follows: precycling hold at 95 °C for 2 min, 42 cycles of 94 °C for 30 s, and 68 °C for 60 s. Next, 10 μL of each PCR product together with 1 μL of the post-stain dye (Midori Green direct) was loaded into 4% agarose gel that was run at 50 V for 50 min. The mutation rate was calculated by determining the sifted band (termed homoduplex and heteroduplex) as compared to control. The resultant PCR products derived from the mixed DNA from three randomly selected mutants from each phenotype class (type I, type II, type III, and type IV) of either TALEN (47-type and 63-type) and three control groups were sub-cloned separately into the vector as described above. Finally, five clones for each of the selected mutant and control groups were subjected to sequencing using the universal primers 'M13_F and M13_R' (Table 1). Genotype analysis was repeated twice (mutant selection, DNA extraction, PCR amplification from mixed DNA of corresponding groups, gel electrophoresis, subcloning, colony selection, and genome sequencing). The sequences obtained were aligned to the wild-type sequence and analyzed using the online software ClustalW-DDBJ version 1.83 (http://clustalw.ddbj.nig.ac.jp/, accessed on 03 December 2021) and CLC sequence viewer 8.0.

As TALENs predominantly induce small deletions at an endogenous locus [50,51], we also analyzed HMA assay based on phenotype classes, for which, we selected embryos derived from the highly active TALEN target B of the scaffold, +136/+63. The resulting PCR amplicons from the mixed DNA (pooled sample of three) of each phenotype class and the control were separated on a post-stain 4% agarose gel in the presence of the 50 bp DNA ladder (NEB, USA).

### 2.8. Histology

TALEN-induced mutants and control embryos at 3 dpf and skin tissue from 60 dpf embryos were collected and fixed in Bouin's solution overnight, which were then dehydrated by means of graded ethanol (50%, 70%, 90%, and 100%) and finally embedded in paraffin. Paraffin sections were prepared at 5μm thickness using a Leica RM2255 microtome (Leica Biosystems Inc. Japan). Sections were stained with hematoxylin and eosin solution followed by cover slipping and imaging with an upright microscope 'BX53' (Olympus). Images were captured using a DP74 CCD camera (Olympus, Japan).

### 2.9. Statistical Analysis

All data are expressed as mean ± SEM. Statistical significance was ascertained using Tukey–Kramer test, with $p < 0.005$ indicating a significant difference between TALEN and control.

## 3. Results

### 3.1. Cloning and Characterization of Kawakawa slc24a5

The kawakawa gene ka-*slc24a5*, which comprises 9 exons and 8 introns and spans across more than 5.7 kb, was analyzed and annotated after extracting the sequence from the whole genome sequence, followed by de novo assembly (Figure 1a). The full-length of the ka-*slc24a5* cDNA sequence (Figure S1) was further confirmed from the retina by RACE PCR. The ka-*slc24a5* cDNA is 2108 bp in length and contains a 113 bp 5'untranslated region (UTR) and 406 bp 3'UTR. There are three potential polyadenylation signals at 1766–1771 bp, 1902–1907 bp, and 2071–2076 bp. The amino acid (AA) sequence of ka-*slc24a5* revealed that the isolated cDNA encoded a 513-AA sequence. It shared 96% homology with that of bluefin tuna and 55–71% with other vertebrates (zebrafish, chicken, frog, wall lizard, and mouse) (Figure S3).

### 3.2. In Vitro Activity of Each TALEN by SSA Assay

An in vitro SSA assay was done to evaluate the activity of the TALENs, ka-*slc24a5*-47A, ka-*slc24a5*-63A, ka-*slc24a5*-47B, and ka-*slc24a5*-, respectively, and determine if the TALENs can induce mutations in the target sequence. Negative controls transfected with each TALEN and reporter vector showed similar activity when compared to those transfected with ZFN and reporter vector, while cells transfected with each TALEN and the associated reporter vector showed much higher activity compared to that of the controls, except for ka-*slc24a5*-47A (Figure 2). In vitro activity of ka-*slc24a5*-47B and 63B showed similar activity.

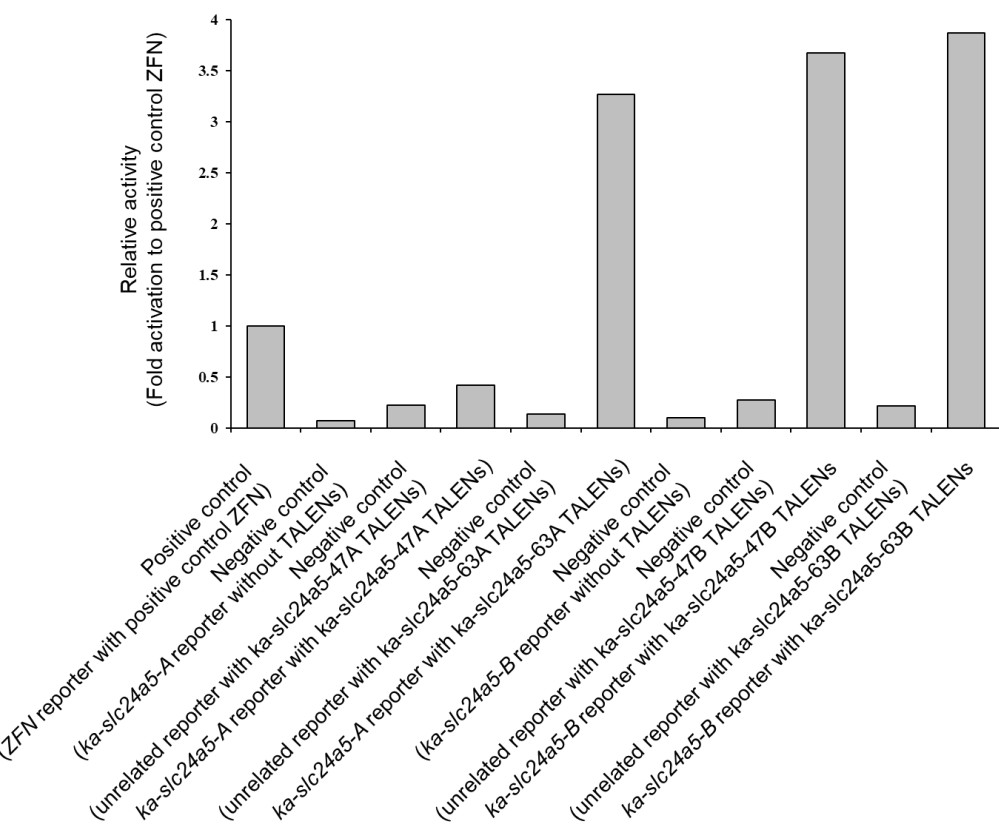

**Figure 2.** Activity evaluation of ka-*slc24a5* TALENs by SSA assay. TALEN plasmids and the corresponding reporter plasmid, TALEN plasmids and unrelated reporter plasmid, or the reporter plasmid without TALEN plasmids were transfected into HEK293T cells. Fold activation scores against the positive control ZFN (pSTL-ZFA36 with the corresponding reporter) were calculated after the luciferase assay.

### 3.3. Phenotypic Change, Survival, and Abnormal Rate after Injecting Each TALENs RNA

We found that active TALEN (ka-*slc24a5*-63B; confirmed by SSA assay) has a tendency to be toxic and affects the survival rate of embryos after injection (with 32% of the injected embryos showing abnormal development with a dwarf or bent body (Figure S4)), whereas TALEN with lower levels of activity (ka-*slc24a5*-47A) is associated with comparatively higher survival rate with no abnormality in the developing embryo (Figures 2 and 3 and Table 2).

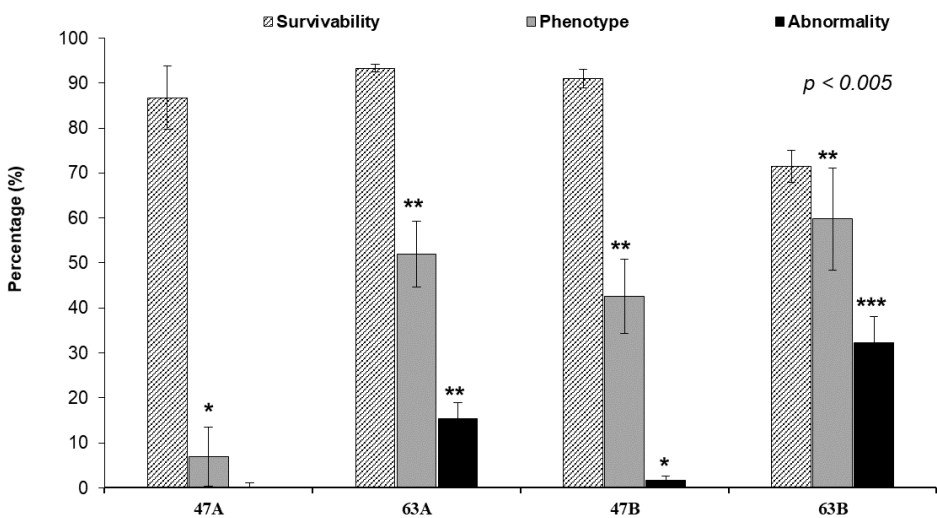

**Figure 3.** Bar graph comparing the efficiency of *slc24a5* TALENs (activity, phenotype, and toxicity) in two different target loci (A and B) of two selected TALE scaffolds (+153/+47 and +136/+63). Data are mean ± SEM (standard error of the mean) for all three parameters (Survivability, Phenotype, and Abnormality) for each TALEN target, with three replicates. Asterisks indicate significant differences ($p < 0.005$) between each TALEN target.

**Table 2.** Survival rate of kawakawa embryos injected with TALENs. Numbers and percentages of embryos surviving during normal and abnormal development at 12 h post fertilization.

| | Experimental Group | Number of Embryo | | | |
|---|---|---|---|---|---|
| | | Total | Survived (%) | Normal Development (%) | Abnormal Development (%) 'Type IV' |
| *slc24a5* 47A | Control | 270 | 242 (89.6) | 242 (100.0) | 0 (00.0) |
| | Injected | 78 | 68 (87.2) | 68 (100.0) | 0 (00.0) |
| *slc24a5* 63A | Control | 149 | 142 (95.3) | 142 (100.0) | 0 (00.0) |
| | Injected | 140 | 130 (92.9) | 110 (084.6) | 20 (15.4) |
| *slc24a5* 47B | Control | 113 | 108 (95.6) | 113 (100.0) | 0 (00.0) |
| | Injected | 67 | 61 (91.0) | 60 (098.4) | 1 (01.6) |
| *slc24a5* 63B | Control | 304 | 285 (93.8) | 285 (100.0) | 0 (00.0) |
| | Injected | 122 | 87 (71.3) | 59 (067.8) | 28 (32.2) |

The control kawakawa larvae showed clear melanin pigmentation in the RPE and skin melanophores located along the ventral contour along the anterior-posterior axis of the body at 72 hpf (Figure 4a). This pigmentation pattern continued to be seen in the larvae until 240 hpf (Figure 4b,c). At 240 hpf, the dorsal view of the control larvae showed a dotted melanin pigmentation between the eyes (Figure 4c). There was an obvious change in the pigmentation pattern in most *slc24a5* mutants after 72 hpf, with the mosaic pattern becoming visible in the RPE at 72 hpf (Figure 4d). However, this pattern became less obvious after 120 hpf and the mutants showed a lower level of melanin pigmentation on the ventral side when compared to the control larvae. At 240 hpf, the dotted pigmentation was less on the head as well in the mutants (Figure 4f). These changes in the pigmentation pattern in the mutants were then classified based on the extent of phenotypic change (Figure 5, and Table S1). The mutants that did not show an obvious change in pigmentation

of the RPE and ventral side of the body when compared to the controls were classified as Type I (low level change; Figure 5a,b); those with a low level of pigmentation in the RPE as Type II (moderate change; Figure 5c); those with pigmentation on the ventral side in addition to the mosaic pattern in the RPE as Type III (severe change; Figure 5d); abnormal development with change in pigmentation pattern as Type IV (Figures 5e and S4).

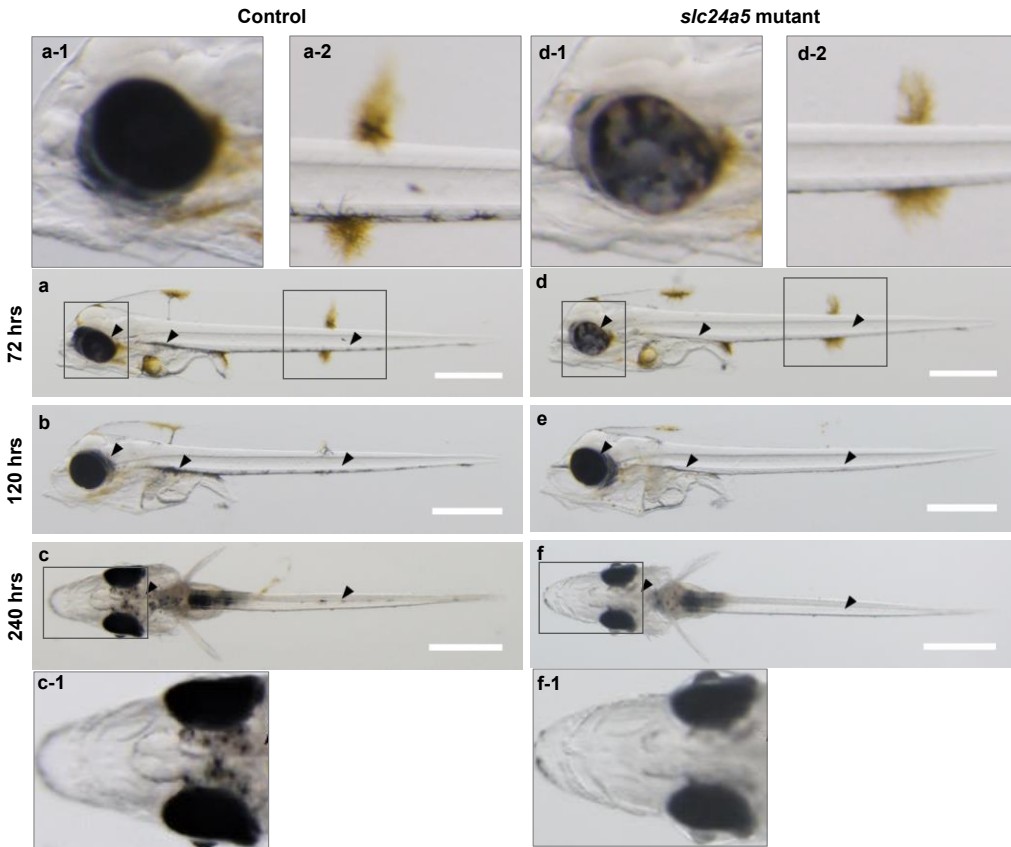

**Figure 4.** Representative images of kawakawa, *Euthynnus affinis*. Control larva (**a**–**c**) and *slc24a5* mutant (**d**–**f**) at 72, 120, and 240 h post fertilization (hpf), respectively. Lateral (**a**,**b**,**d**,**e**) and dorsal views (**c**,**f**) are shown. The regions shown in a and d, are enlarged in a-1, a-2 and b-1, b-2, respectively, while regions shown in c and f are enlarged in c-1 and f-1 respectively. In intact larvae, melanin pigmentations were clearly observed at the eyes and ventral side of body at 72 hpf and on the dorsal side of head at 240 hpf. In the *slc24a5* mutant, a mosaic pigmentation pattern was observed at 72 hpf and a lower level of pigmentation on the ventral side of body and head at 120 and 240 hpf. Pigmentation is shown by means of arrowheads. Scale: 1 mm.

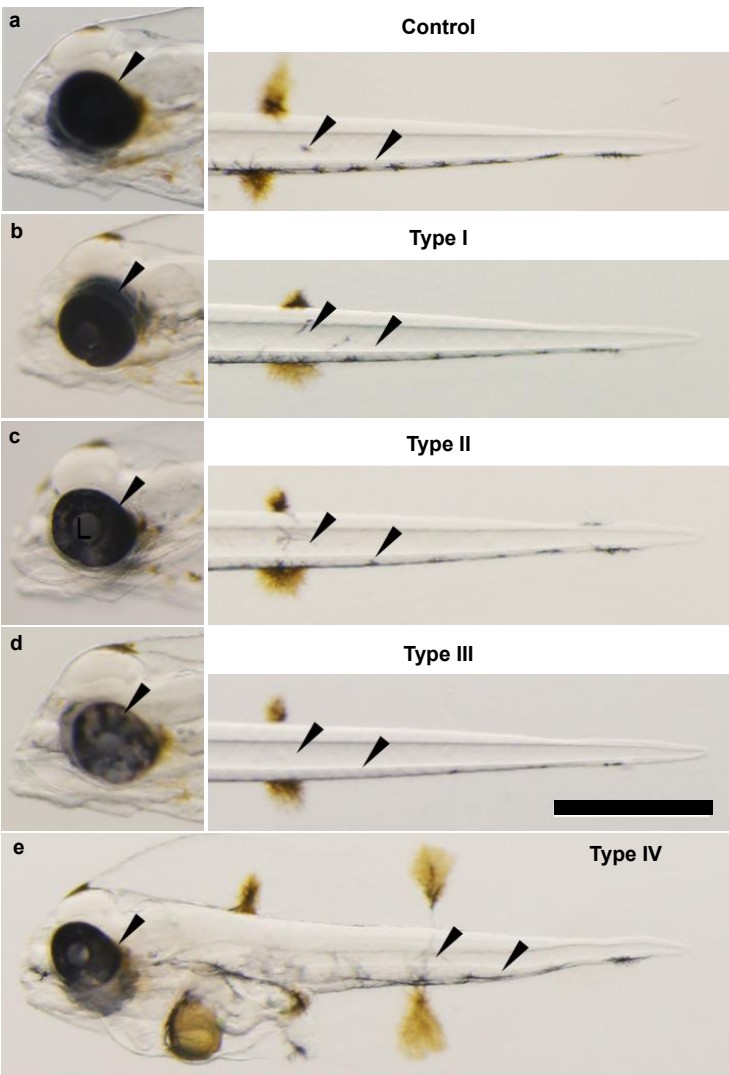

**Figure 5.** Phenotypic classification of pigment patterns in the *slc24a5* mutant at 72 hpf. (**a**) Melanin pigmentation in a control larva; (**b**) Type I: mutant with normal body development and pigmentation pattern similar to control, but is genotypically different; (**c**) Type II: mutant with normal body development, with moderate phenotype (mosaic eye); (**d**) Type III: mutant with severe phenotype (mosaic eye, low level of body pigmentation) with normal growth; and (**e**) Type IV: mutant having abnormal body development (short and twisted body, comparatively large head, among other abnormalities). Scale: 1 mm.

Histological examination of the eyes confirmed the low level of melanin pigmentation in the *slc24a5* mutants (Figure 6). A cross section of the head through the eyes showed a darker layer of RPE in the control larvae (Figure 6a) when compared to the *slc24a5* mutants (Figure 6c). A dark and dense pigmented cell layer of RPE in the outer most layer of the retina attached to photoreceptor layer (PRL) was further confirmed in the enlarged picture of the control larvae (Figure 6b) as opposed to the less pigmented cell layer in the *slc24a5* mutant at 72 hpf (Figure 6b). However, there were no remarkable differences observed in the structure of the lens, optic nerve, inner nuclear layer, and inner plexiform layer between the control and experimental groups. A rather light mosaic pattern was also seen in the RPE of the mutants when compared to the control embryos at 120 hpf (data not shown). In addition, the pattern of pigmentation in the dorsal skin at 60 dpf clearly showed fewer number of melanophores and a low level of melanin deposition in the mutants (Figure 7d–f) when compared to the control larvae (Figure 7a–c). Although chromatophores (Ch) are pigment containing cells, there was color variation observed in the dorsal skin of mutants at 60 dpf due to low density of Ch. The high-density Ch in the control

larvae (Figure 7c) and low-density Ch in the mutants (Figure 7f) were observed to be between two layers, namely the outermost skin layer (epidermis; EP) and the dermis (D).

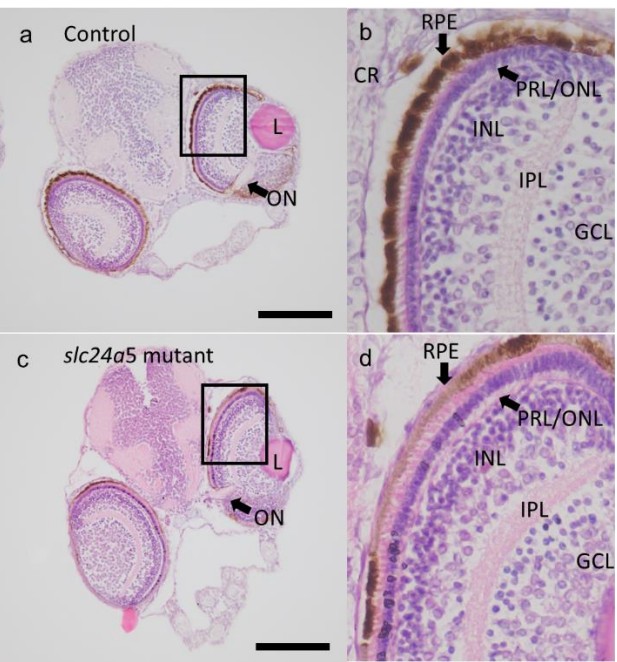

**Figure 6.** Histological section of kawakawa eyes in the *slc24a5* mutant at 72 hpf. A cross section of the head in control larva (**a**,**b**) and *slc24a5* mutant larva (**c**,**d**). The regions shown in a and c are enlarged in b and d, respectively. Control larva showed completely dark RPE with melanin pigmentation, while much lighter RPE without melanin pigmentation was seen in the *slc24a5* mutant. RPE: retinal pigment epithelium; PRL/ONL: photoreceptor layer/outer nuclear layer; INL: inner nuclear layer; IPL: inner plexiform layer; GCL: ganglion cell layer; L: lens, ON: optic nerve; CR: choroid rete. Scale 100 μm.

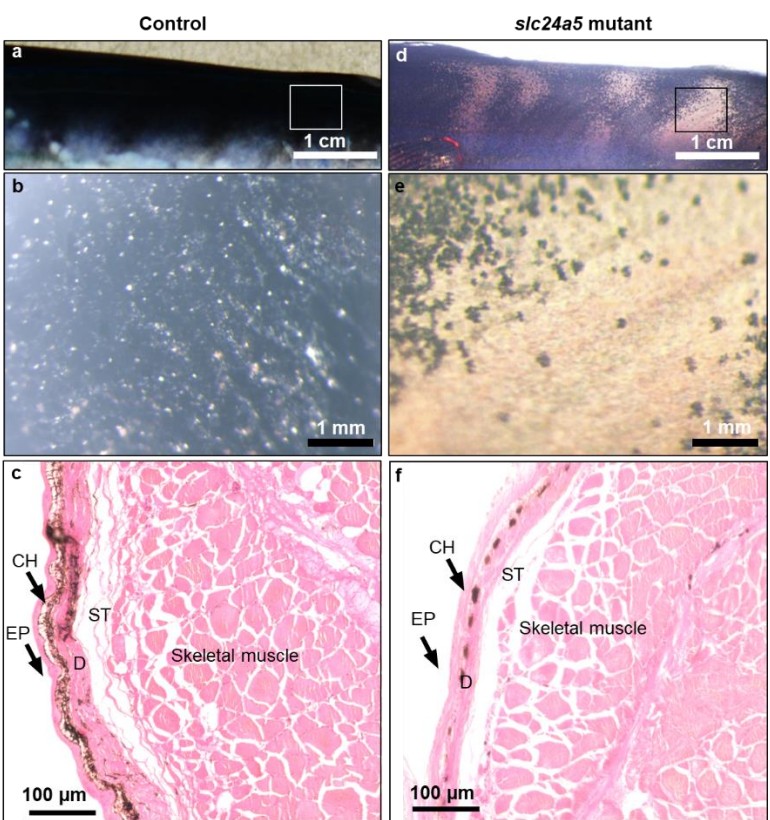

**Figure 7.** Melanin pigmentation of dorsal skin of kawakawa at 60 dpf. Pigmentation phenotype of the dorsal skin of a control juvenile (**a**) and *slc24a5* mutant (**d**). Squared regions in a and d are enlarged in (**b**,**e**), respectively. Images of (**c**,**f**) were derived from a histological section after HE-staining of the dorsal skin from the control and *slc24a5* mutant, respectively. Lower level of pigmentation on the dorsal skin of the *slc24a5* mutant was further confirmed in the histological section, showing fewer melanophores under the epidermis. CH: chromatophore; EP: epidermis; D: dermis; ST: stratum.

### 3.4. Heteroduplex Mobility Assay (HMA) and Genotype

The PCR amplified product obtained from the mixed DNA of the TALEN target site was analyzed using HMA. Most of the indels occurred in the somatic cells of TALEN-injected F0 founder during embryogenesis. We found multiple DNA bands of fast migrating homoduplex and slow migrating heteroduplex from the TALEN-injected embryos and a single linear sharp band from the control in the agarose gel (data not shown). The HMA result obtained from the selected sample based on phenotype classes confirmed the different patterns of the multiple DNA bands. There are two heteroduplex bands from type I, a single heteroduplex band from type II, six different heteroduplex bands from type III, and two heteroduplex bands with cleavage fragments from type IV mutants (Figure 8). We also examined if HMA could be used to genotype F0 embryos by DNA sequencing (described in Section 2.7). Sequencing of randomly selected fish indicated that TALENs predominately induce mutations by way of small indels (Table 3). The TALEN target showed that indels made up 60–100% of the mutations in either phenotype group of scaffolds (+157/+47 and +136/+63), while single nucleotide changes contributed to 27% of the mutations in ka-*slc24a5*-47A. Thus, the multiple HMA profile correlated with the efficiency of TALEN-induced mutations (substitution, addition, and deletion) subjected to activity confirmed from an in vitro assay (Figures 2 and 3 and Table 3).

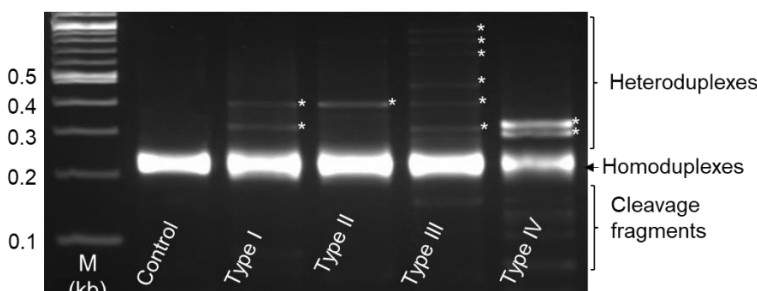

**Figure 8.** Heteroduplex detection by PCR followed by gel electrophoresis. Multiple homoduplex and heteroduplex bands (asterisks) were present in PCR products from each classified type (phenotypic classes within *slc24a5* TALEN-injected embryos), whereas a single sharp band was present for the control embryo (no TALEN injection). The molecular size of the marker (50 base pair DNA ladder) is shown on the left. M: molecular marker; kb: kilobase.

**Table 3.** Genotype and appearance ratio in F0 *slc24a5* genetically modified by TALEN in kawakawa.

| Target group | | Sequences (5'-3') | Genotype (%) |
|---|---|---|---|
| **Group A** | Wild | tctgcagaaaaagaaaagaAAAGATTTTATTCCT<u>tacttcctgggatttgta</u> | (10/10) |
| *slc24a5*, 47A | Type II | TCTGCAGAAAAAGAAAAGAAAAGAWTTTATTCCTTACTTCCTGGGATTT-GTA | 4/15 (27%) |
| *slc24a5*, 63A | Type II | TCTGCAGAAAAAGAAAAGAWAARWTTTTATTCCTTACTTCCTGGGATTT-GTA | 7/15 (47%) |
| | | TCTGCAGAAAAAGAAAAGAAAAR----------TTACTTCCTGGGATTTGTA | |
| **Group B** | Wild | tcctgggatttgtactattTTTATATTGCACRGYC<u>catcttgtatcattcaca</u> | (10/10) |
| *slc24a5*, 47B | Type I | TCCTGGGATTTGTACTATTTTTWccaTATTGCAYRGYCCATCTT-GTATCATTCACA | 7/10 (70%) |
| | Type I | TCCTGGGATTTGTACTATTTT-------CACG---CATCTTGTATCATTCACA | |
| | Type II | TCCTGGGATTTGTACTATTTTTWTATWGCtatttACGGTCCATCTT-GTATCATTCACA | 9/10 (90%) |
| | Type II | TCCTGGGATTTGTACTATT----------------CATCTTGTATCATTCACA | |
| | Type III | TCCTGGGATTTGTACTATTTTTTATActatttTTGCACRGYCCATCTT-GTATCATTCACA | 10/10 (100%) |
| | Type III | TCCTGGGATTTGTACTATT------------------TCTTGTATCATTCACA | |
| *slc24a5*, 63B | Type I | TCCTGGGATTTGTACTATTTTTWTATTGCACRGYCCATCTT-GTATCATTCACA | 6/10 (60%) |
| | Type I | TCCTGGGATTTGTACTATTTT-------------CCATCTTGTATCATTCACA | |
| | Type II | TCCTGGGATTTGTACTATTTTTAYDWWGCACSGTCCATCTT-GTATCATTCACA | 7/10 (70%) |
| | Type II | TCCTGGGATTTGTACTATT---------------CATCTTGTATCATTCACA | |
| | Type III | TCCTGGGATTTGTACTATT---------------CATCTTGTATCATTCACA | 7/10 (70%) |
| | | {(x) left TALEN, (<u>x</u>) right TALEN}　　　　[(X)substitution, (x)insertion, (-) deletion] | |
| | | D=A/G/T; R=A/G; S=G/C; W=A/T; Y=C/T | |

*3.5. Analysis of slc24a5 Transcripts by RT-PCR and Quantitative Real-Time PCR*

Expression of kawakawa *slc24a5* in embryos at different stages of early development and in different adult organs was carried out by RT-PCR. A set of specific primers from the reference gene '*β-actin*' was used as a positive control for both groups (different stages of developing embryos and adult tissues). All experimental samples showed a unique and uniform single band from the reference gene at the target size of 0.5 kb (Figure 9a,b). Tissue-specific *slc24a5* RNA was expressed in all selected tissues, with a relatively strong expression in the eye, followed by skin, and a weak band in other tissues (gut, liver, muscle, heart, kidney, pylorus, gill, gonad, and brain; Figure 9b). However, *slc24a5* expression was initiated in certain embryos only at 2 dpf (48 hpf) and later (Figure 9a). There was no *slc24a5* RNA expression found in the early stages of the embryos (2-cell, 128-cell, 1K-cell, epiboly, and somite stages). In addition, a quantitative real-time PCR confirmed the lower expression of *slc24a5* RNA in the early embryonic stages of 13 and 24 hpf, whereas the control group showed a remarkable increase at age 72 hpf followed by a sudden decrease at 120 hpf. The level of *slc24a5* RNA expression in control embryos was significantly different from that of the TALEN-induced mutant only at 72 hpf. The TALEN induced mutants showed a significantly low expression of *slc24a5* RNA (loss of function) at 72 hpf when compared to the control larvae (Figure 10).

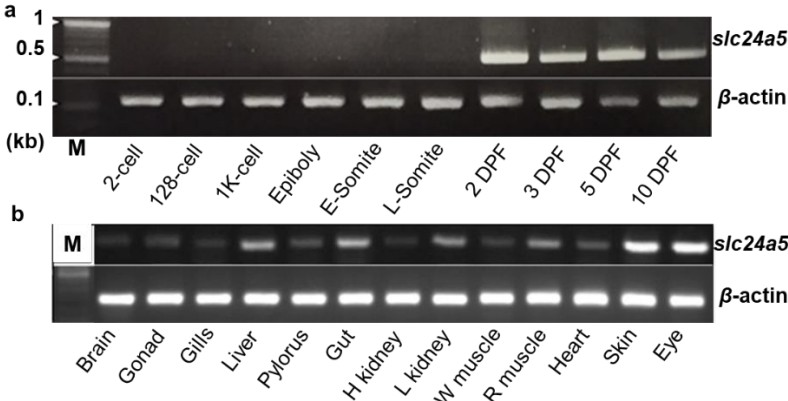

**Figure 9.** *slc24a5* RNA expression in kawakawa. (**a**) Stage dependent mRNA expression (embryos at different stages of developmental time point). (**b**) Tissue specific mRNA expression (Adult, N-6). RT-PCR was done on cDNA derived from adult tissues (brain, gonad, gills, liver, pylorus, gut, kidney, muscle, heart, skin, and eye) and embryos at different developmental time points (0.1–10 dpf) using oligonucleotide primers (TD_F and TD_R results in expected amplicon of 501 bp) that flanked the specific *slc24a5* in kawakawa. Positive controls '*β-actin*' transcripts (β-act-F1 and β-act-R1 results in amplicon of 105 bp) were present in all selected samples including different tissues from adults, and embryos at different stages of developmental time point. The molecular size of the marker (50 base pair DNA ladder) is shown on the left. M: molecular marker; kb: kilobase; dpf: days post fertilization.

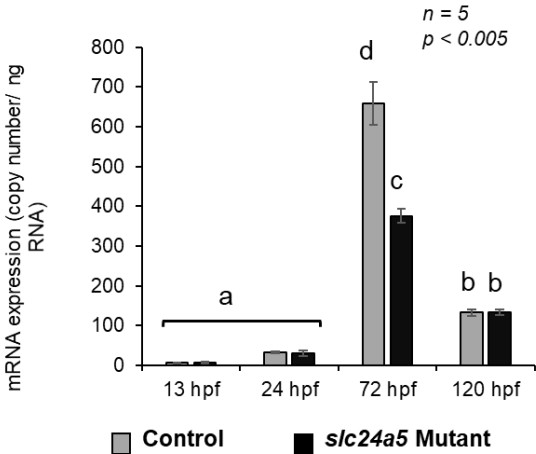

**Figure 10.** *slc24a5* mRNA expression analysis in control and *slc24a5* mutants at different developmental time points (13-120 hpf) of kawakawa embryos using quantitative real-time PCR. Data are mean ± SEM from four developmental stages, with three replicates each. Letters (a–d) on the top of the bar indicate significant differences (*p* < 0.005) between control and the TALEN-induced mutant *slc24a5* expression. The absolute quantification of the *slc24a5* transcript in the TALEN mutant was significantly different from that of the control embryos only at 72 hpf. hpf: hours post fertilization.

## 4. Discussion

Skin color variations are widely reported in various animal phyla [30,31,36,41,42,44,52]. The levels of skin color differences vary among species as part of a fitness strategy. In animals, the mechanism of dark pigmentation in the dorsal skin and light pigmentation in the ventral skin is thought to protect vital issues from harmful radiation [53]. The mackerel tuna, kawakawa (*Euthynnus affinis*) also follows the same pigmentation pattern with the dorsal skin uniformly covered with a dark pigment and the ventral side with a light, pale belly. There have been no reports of any variation in skin color in any tuna species, including kawakawa. Here, we demonstrated the disruption of the gene, *slc24a5*, using Platinum TALENs in kawakawa, marking the use of genome

editing technology to study color variation in tuna species, for the first time. Mutant ka-wakawa exhibit lower levels of melanin pigmentation in RPE during early development and pigment-less patches on the body, which has also been observed in zebrafish [36,54] and frog [30]. Therefore, it seems reasonable to conclude that *slc24a5* is responsible for the golden phenotype in kawakawa, as in other animals, such as zebrafish.

The golden phenotype appears only in homozygous *slc24a5* mutants since it is a recessive trait [30]. In our study, it was not clear that there would be phenotypic changes in the F0 kawakawa after genome editing of *slc24a5* gene as fish with the wild-type phenotype was used for this experiment. By disrupting *slc24a5* using TALENs, the order of the melanin deposition in eye and skin of the mutants appears to have been changed, when compared to the control. Microscopic observations and histological analysis clearly showed that less pigmented tissues appeared in RPE and skin at 72 hpf in the TALEN-injected groups. Although the levels of pigmentation in the RPE were different among individual mutants, there were no completely pigment-less mutants observed in kawa-kawa, as in zebrafish [30]. This could be due to the timing of injecting TALEN RNAs into eggs. We collected the fertilized eggs from sea net cages after visual confirmation of spawning. Therefore, the injection might have been early enough to induce complete loss of pigmentation in kawakawa. After 72 hpf, the pigmentation in RPE was gradually restored and there was no significant difference in RPE pigmentation between control and the mutant under stereomicroscope observations. These changes of pigmentation pattern were comparable to those in zebrafish [36]. However, there was a loss of melanin in the skin of mutants during the entire period of the experiment. Pigment-less patches that appeared on the body of juvenile mutant kawakawa indicated that disruption of *slc24a5* gene results in lighter body color in this species.

Embryonic expressions of *slc24a5* RNA were determined using RT-PCR. The *slc24a5* RNA were first detected in the larvae at 2 dpf and it was continuously expressed up to 10 dpf, which was the latest stage examined in this study, suggesting that there was no maternal RNA expression in early stages of development in kawakawa and confirming that the expression of *slc24a5* in early development is only zygotic. Therefore, changes of *slc24a5* RNA in the mutants are the result of genome editing. Moreover, real-time PCR showed that *slc24a5* RNA expression in normal development is initiated at 48 hpf in kawakawa embryos and undergoes a remarkable increase at 72 hpf. This differential *slc24a5* expression as embryonic development progressed correlated with the deposition of brown pigments at 2 dpf and pigmentation in the RPE, with slightly dark body pigmentation at 3 dpf. In addition, the tissue-specific RNA expression using semi quantitative RT-PCR confirmed the strong DNA band in those tissues, which had a high level of melanin deposition. In other words, the expression of *slc24a5* was highest in retinal tissue (eye) followed by skin, as observed in chicken [55] and mouse [36,37].

Together with measuring the expression of *slc24a5* in early development, real-time PCR analysis was also done to examine the *slc24a5* RNA expression in the mutants. The primers used for real-time PCR were designed to amplify both the target sequences of TALENs, ka-*slc24a5*A and ka-*slc24a5*B. The intercalation method for real-time PCR used in this study detects the amount of double stranded DNA by means of fluorescence. Therefore, PCR products contained not only intact *slc24a5* cDNA but also any type of cDNA amplified from the modified slc24a5 gene. Despite this methodological disadvantage, we detected a significant difference of *slc24a5* RNA expression in TALEN mutants with low expression when compared to the control at 72 hpf. This finding indicates that functional protein levels of *slc24a5* were lower in the mutants, resulting in low levels of pigmentation in the mutant larvae.

In this study, we designed two scaffolds, +153/+47 and +136/+63, at the same target sequences, to compare the activity in vivo. Unfortunately, target A of scaffold +153/+47 showed rather low level of activity in the SSA assay and low genome editing activity in vivo. Target B showed a similar activity between the two scaffolds in the SSA assay that was also confirmed in vivo between these two TALENs. However, it appears that the

activities of TALENs directly correlated with toxicity, as the side-effect of highly active TALENs (both targets of scaffold +136/+63) was higher when compared to the less active TALEN (target A of scaffold +153/+47). Furthermore, one TALEN pair, ka-*slc24a5*-47B, generating >50% mutant alleles in individual embryos suggested that the efficiency varied depending on the TALEN pair injected, comparable to the results of different TALEN pairs in other model organisms such as zebrafish [17,56].

Artificial production of seedlings, especially in tuna, is difficult due to the high mortality rate when compared to other aquaculture species. In this study, it was difficult to rear the kawakawa to adulthood for reasons other than the effects of the TALEN injection, which did not last beyond early stages of development. Although mortality of artificially reared tuna juveniles in sea cage or tanks is typically high due to injuries from collisions with the walls as the panicked fish attempt to escape by swimming at high speeds, platinum TALEN-induced mutants of the Pacific bluefin tuna embryos were relatively calm and swam slowly, resulting in comparatively higher survival [57]. In case of kawakawa, a small-scale rearing system for experiments has been established in our laboratory where the tanks are larger in size and the fish are moved to net cages when the fish reach a certain size. However, the F0 mutants did not survive for more than 3 months, due to the high rates of cannibalism in the early-stage larvae and certain limitations in the rearing conditions for GMOs.

Any mutation in the F0 needs to be in the germline rather than somatic tissue such as the skin, for it to be passed on to the offspring. In addition, there is a two-step mating process necessary to obtain *slc24a5* homozygote offspring [26], for which, it is necessary to establish the genome edited founder line (F1 generation) that can also be used for further research.

We also developed an efficient method for the detection of TALEN-induced mutations using the HMA assay. This approach consists of a simple PCR amplification of DNA fragment containing the TALEN targeted sequence, followed by gel electrophoretic analysis. Multiple patterns of heteroduplex band point to specific levels of mutation efficiency of TALEN-induced embryos, based on their phenotype classes. The DSB repair-mediated deletions are mainly a result of the three DSB repair pathways, classical non-homologous end joining (cNHEJ), microhomology-mediated end joining (MMEJ), and SSA [58]. During the genotype analysis, we found that some pairs of TALEN-induced specific patterns of mutation, including deletion (10, 14, 16, and 18 bp), addition (3, 5, and 6 bp), and substitution of nucleotides between regions of homologous sequence near the spacer region of these TALENs are consistent with the results observed in medaka [23], suggesting that the DSB repair was likely to be done by MMEJ.

In conclusion, we identified a single *slc24a5* gene in kawakawa, with a molecular mass of 56.82 kDa, that shared 96% homology with that of bluefin tuna *slc24a5*. The aim of the present study was to establish a genome editing technique using TALENs in a model marine organism that was suitable for studying tuna species. In our research design of this study, efficient phenotypic changes and small genetic modifications by way of substitution, insertion, and deletion around the target site, by disruption of the *slc24a5* gene confirmed that TALENs predominantly induce small mutations with major changes in phenotype. We also found that the diversification of the pigmentation pattern in kawakawa is closely related to changes in the transcriptional regulation of black pigmentation. In summary, we developed a comprehensive experimental system for kawakawa, and presented results that provide interesting and useful insights into efficient genome editing of kawakawa using TALENs that can be used to create mutations at specific loci. These studies of genome modification in kawakawa hold significant promise for efficient aquaculture production by improving growth, survival, and disease resistance.

**Supplementary Materials:** The following are available online at www.mdpi.com/article/10.3390/jmse9121378/s1. Figure S1. cDNA and genome structure of *slc24a5* in kawakawa, *Euthynnus affinis*. Figure S2. Dendrogram showing *slc24a5* relationships between kawakawa with those

of other selected species based on comparisons of their polypeptide sequence deduced from cDNA. Numbers adjacent to each node represent bootstrap values (100 bootstraps replicates). Reference amino acid sequence of *slc24a5* used in this study from different species, and their GenBank accession numbers are showing on the right-hand side. Figure S3. Deduced amino acid sequence alignment of kawakawa *slc24a5* (MH_182681.1) with other species; Tuna (Atlantic bluefin tuna, unpublished data), Zebrafish (NP_001025451), Chicken (NP_001033586.2), Wall Lizard (XP_028561132.1), Mouse (NP_778199.2), and Frog (XP_002936612.2). Identical residues in each species of peptides are marked in white font on a black background while less similar peptide shown in black font on a gray background. Non-identical peptides are shown in black font on a white background. Figure S4. *slc24a5* TALEN induced mutants and their body development (sign of toxicity): a mutant with normal development at 72 hpf (a); a dwarf body at 72 hpf (b), and a twisted body at 120 hpf (c). Scale: 1 mm. Figure S5. A complete nucleotide sequence along with translated amino acid sequence of kawakawa *beta-actin* gene. The number on left side represents the corresponding position of the amino acid and cDNA residues, respectively. x—start codon; x—stop codon; x—3′ untranslated region. Table S1. Phenotype and appearance ratio in F0 *slc24a5* genetically modified by TALEN.

**Author Contributions:** Experiment design; D.P. and R.G. Platinum TALEN construction; D.P., R.G., T.S. (Tetsushi Sakuma), T.Y., Y.K., K.G. and M.M. TALEN RNA injection and sample collection; D.P., R.G., T.M. and T.S. (Taiju Saito). Conduction of experiments; D.P. and R.G. Manuscript writing; D.P. and R.G. All authors contributed to the final version of the manuscript. All authors have read and agreed to the published version of the manuscript.

**Funding:** This study was supported by JSPS KAKENHI Grant number JP17H03873; Cross-ministerial Strategic Innovation Promotion Program(SIP), "Technologies for creating next-generation agriculture, forestry and fisheries" (funding agency: Bio-oriented Technology Research Advancement Institution); Regional innovation ecosystem formation program "The Ehime Fisheries Innovation Ecosystem Initiative (FY2017-2021)" supported by ministry of education, culture, sports, science and technology Japan (MEXT).

**Institutional Review Board Statement:** The animal study was reviewed and approved by Animal Care Committee of Ehime University.

**Informed Consent Statement:** Not applicable.

**Data Availability Statement:** Not applicable.

**Acknowledgments:** The authors thank entire team of South Ehime Fisheries Research Center for their kind support to complete this experiment; especially who helped with rearing of fish, Eitaro Sawayama, Taketo Hayakawa, and Ryoko Kawata.

**Conflicts of Interest:** The authors declare no conflict of interest. The funders had no role in the design of the study; in the collection, analyses, or interpretation of data; in the writing of the manuscript, or in the decision to publish the results.

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
