# Peer review of "TALEN-Mediated Gene Editing of slc24a5 (Solute Carrier Family 24, Member 5) in Kawakawa, Euthynnus affinis"

_jmse, doi:10.3390/jmse9121378_

Round 1
Reviewer 1 Report
The following suggestions are to be included before the acceptance of
TALEN-mediated gene editing of slc24a5 (solute carrier family 24, 2 member 5) in kawakawa, Euthynnus affinis
Abstract: It is written well it acts as a short trailer of the entire paper containing every little detail of the article. Entirely a brief summary.
Introduction:
- In the 2nd Paragraph (Sentence no 38 to 46) has information about the fish, its habitat, and its transformation as aquaculture species, but no reference is mentioned.
- Statement related to Talen being more specific than CRISPR/Cas does not have any reference, so, is it the interpretation of the author of this paper? (Sentence line 61-63)
- It’s unclear can maybe require elaboration(sentence lines 73-76).
- Materials and Methods:
2.1 three females and three males of adult kawakawa were sacrificed: Whether it scarificed by using anaesthetics?
Did you get permission from the Institute Bioethics Committee to carry out GMOs? (Mentioned approval number)
Experimental animals and tissue collection (sentence lines 123 & 124) seem to be repeated under the section.
"Embryos at the 1‒2 cell stage were placed on a 1% agar-coated Petri dish filled with 50% filtered seawater, and then microinjected with TALEN mRNA at different doses.(50 ng, 100 ng and 150 ng) each of the left and right TALENs" No proper explanation in the result section why different doses of TALEN mRNA microinjected to the embryos and was its effect. (Materials method; sentence line 226-228)
Figures 3 and 4 - Couldn’t differentiate much between the control and slc245a mutant of pigment pattern development. Higher magnifications (TEM/ confocal microscopy) could be used to know the melanin pigmentations (larval melanophores & chromatoblasts) in sentence lines 325 & 332).
Results and Discussion are written well with recent references
As per the S/W search engine, report.net overall in entire paper appx 6% of plagiarism was detected.
Even no study on genome editing of slc24a5 in Euthynnus affinis was found proves the paper's uniqueness.

Reviewer 2 Report
This manuscript presented methodolgy and results of genome editing using TALENTs for slc24a5 in Kwakwa, one of the tuna fish in Indo Pacific. Although only mosaic fish were obtained, this manuscript showed some evidence of function of slc24a5 associating with melanin pigment formationn, demonstrating the application potentials of TALENT-mediated gene editing in aquaculture and genetic improvement.
Comment: In Fig 9, "n=5" for what?
